# Neural Bases of Affect-Based Impulsivity: A Decision Neuroscience Account

**REVIEWS**

]u[ ubiquity press

**ALISON M. SCHREIBER** (iD)

**MICHAEL N. HALLQUIST** (iD)

*Author affiliations can be found in the back matter of this article

## ABSTRACT

Affect-based impulsivity describes the tendency to behave impulsively while experiencing negative or positive affective states. In the context of psychiatric disorders, the consequences of affect-based impulsivity can be dire, including suicidal behavior and harmful substance use. Here, we provide a narrative review and articulate a decision neuroscience account of affect-based impulsivity. We focus specifically on how negative emotions alter the balance of Pavlovian and goal-directed decision systems. We consider how negative affect influences *whether* to act, *what* actions to consider, *which* action to select, and *how* vigorously to engage in a selected action. Further, we describe the neural and neuroendocrine bases of these computations. We propose that modulation of norepinephrine and glucocorticoids during negative affective states enhances the pursuit of rewards by reducing goal-directed computations and increasing appetitive Pavlovian computations.

**CORRESPONDING AUTHOR:**
**Alison M. Schreiber**

Department of Psychiatry, University of Kentucky, Lexington, USA

alisonmschreiber@uky.edu

**KEYWORDS:**
impulsivity; emotion; reinforcement learning; decision neuroscience; computational psychiatry; decision-making

**TO CITE THIS ARTICLE:**

Affect-based impulsivity is defined as engaging in impulsive behaviors during an emotional state (Cyders & Smith, 2008). Affect-based impulsivity varies dimensionally in the population, and elevated levels have been observed in several psychiatric illnesses including borderline personality disorder, substance use disorders, bipolar disorder, and bulimia nervosa (Berg et al., 2015; Cyders & Smith, 2008). Longitudinal studies find that affect-based impulsivity predicts the onset and worsening of psychiatric symptoms (Cyders et al., 2010; Cyders & Coskunpinar, 2010; Manasse et al., 2018; VanderVeen et al., 2016). In addition, affect-based impulsivity portends risk for problematic drinking, pathological gambling, and compulsive shopping (Cyders & Smith, 2008). Affect-based impulsivity encompasses impulsive behaviors in response to both positive and negative emotions (Smith et al., 2007), forming a valence-independent factor that is more robustly associated with clinical outcomes than other facets of impulsivity (e.g., sensation seeking; Berg et al., 2015).

Research on the neurocognitive mechanisms of affect-based impulsivity has primarily considered four accounts (Johnson et al., 2020): (1) heightened emotion generation, (2) impaired emotion regulation, (3) risky decision-making, and (4) impaired response inhibition (see Fisher-Fox et al., 2024 for additional alternative accounts). Empirical studies have found little evidence of heightened emotion generation (Amlung et al., 2017; Cyders et al., 2010; Cyders & Coskunpinar, 2010, 2011; Johnson et al., 2017; Owens et al., 2018; Pearlstein et al., 2019; VanderVeen et al., 2016; Wise et al., 2015) and mixed evidence for risky decision-making (Cyders et al., 2010; Cyders & Coskunpinar, 2011; Johnson et al., 2016; Mackillop et al., 2014; MacKillop et al., 2016; Sharma et al., 2014; Wise et al., 2015). Conversely, affect-based impulsivity is reliably associated with the use of less effective emotion regulation strategies and weaker recruitment of brain regions involved in emotion regulation (Albein-Urios et al., 2013, 2014; King et al., 2018). Moreover, affect-based impulsivity is associated with impaired response inhibition (Cyders & Coskunpinar, 2011; Dekker & Johnson, 2018; Johnson et al., 2016; Sharma et al., 2014), particularly in clinical samples (Dekker & Johnson, 2018), and altered neural processing during inhibitory control tasks (Barkley-Levenson et al., 2018; Chester et al., 2016; Tervo-Clemmens et al., 2017; Wilbertz et al., 2014). Studies to date have primarily relied on summary indices of emotion processing or response inhibition (e.g., rate of inhibitory failures), which only provide indirect evidence about lower-level cognitive processes (Gureckis & Love, 2015; Love, 2015) and often have poor psychometric properties (Hedge et al., 2018). More recently, computational approaches have yielded insight into the generative processes that produce behavior (Huys, Guitart-Masip, et al., 2015; Montague et al., 2012), formally bridging behavior and the brain (Love, 2015; Palmeri et al., 2017). Even as research on the neurocognitive underpinnings of affect-based impulsivity has grown over the past 20 years, much less is known about how an emotional state shapes the neurocomputational processes that underpin impulsive decisions.

Here, we present a narrative review of affect-based impulsivity, propose an integrative decision neuroscience account of affect-based impulsivity, and outline a research agenda for testing central components of our account. We organize the review around the core components of our model. Given our focus, we primarily review human research that uses theory-based computational psychiatry methods (Bennett et al., 2019; Maia et al., 2017), as well as preclinical animal neuroscience studies that elucidate corresponding neural pathways (e.g., Cartoni et al., 2016; Glimcher, 2011; Niv et al., 2006). Nonetheless, we appreciate the value of other approaches, including individual differences research (Cyders & Smith, 2008; Fisher-Fox et al., 2024), ecological momentary assessment (Sperry et al., 2021), and cognitive and affective neuroscience (Johnson et al., 2020).

We narrow the scope of our review in a few key ways. First, although trait affect-based impulsivity varies across people (Cyders & Smith, 2008), our account focuses on the within-person effect of emotions on neurocomputational decision processes. That is, how do negative emotions alter neurocomputational processes, relative to that person's baseline? We anticipate that this within-person account can inform our understanding of trait affect-based impulsivity (consistent with personality theories that view traits as a density distribution of states; Fleeson, 2001), since individual differences in computational and neural systems could lead to between-person

Schreiber and Hallquist    **87**
*Computational Psychiatry*

differences (Huys, Guitart-Masip, et al., 2015). Second, impulsivity is not a unitary construct, referring to distinct behavioral tendencies that range from heightened delay discounting to inhibitory control deficits to an impulsive decision-making style (Caswell et al., 2015; Sharma et al., 2014). In our model, we focus on impulsive behaviors that are short-sighted and rash, reflecting a preference for immediate rewards, consistent with original psychological theories on trait affect-based impulsivity (Cyders & Smith, 2008; Smith et al., 2007).

Lastly, we narrow the scope of our account to focus on negative emotions, consistent with research finding distinct neural correlates for negative and positive emotions (Lindquist et al., 2012, 2016). Emotions are often associated with certain action tendencies (N. Frijda et al., 1989; N. H. Frijda, 1987; Izard, 2007; Lang & Bradley, 2013; Moors et al., 2013; Posner et al., 2005), and aversive states often inhibit behavior altogether (McNaughton & Corr, 2004). Given that negative emotions are often associated with inaction and withdrawal (e.g., de Berker et al., 2016), it is potentially counterintuitive that aversive internal states would promote pursuit of appetitive cues.

To explain such phenomena, we describe how emotion-dependent computations affect four aspects of decision-making: 1) whether to act, 2) what actions to consider, 3) which action to take, and 4) how vigorously to act. Although we describe these as sequential yet overlapping computations, we do not mean to imply that each stage involves conscious deliberation. Rather, we view these four aspects as interacting processes that may unfold rapidly and in parallel. We formalize key components of this model into two hypotheses: First, negative affect leads to an increase in circulating glucocorticoids (GCs) and norepinephrine (NE). GCs and NE blunt medial prefrontal cortex (mPFC) functioning (Arnsten, 2009; Schwabe et al., 2012), reducing goal-directed computations that support deliberative reasoning (Gläscher et al., 2010). Second, affect-related increases in GCs sensitize the mesolimbic dopamine (DA) pathway, enhancing the influence of Pavlovian reward cues on behavior (Peciña et al., 2006; P. V. Piazza & Le Moal, 1996). Altogether, this account provides new biological and neurocomputational targets for understanding impulsive decision processes in humans.

## AVERSIVE INTERNAL STATES SHAPE DECISION-MAKING

How might negative emotions enhance the pursuit of rewards? Let us consider an example. Ada and her boyfriend Darius are troubleshooting their TV connection. Ada considers different actions, with the goal of being able watch her favorite reality TV show, The Bachelor. Then, Ada receives a text asking her and Darius to bring a dessert to a potluck (Figure 1a). They disagree over what to bring, and this disagreement escalates into an argument. Ada notices a shift in her internal state (e.g., racing heart) and labels this state "shame". She finds this state aversive and is now focused on reducing these feelings of shame. She decides to go on a walk, hoping that she will begin to feel calmer (Figure 1b). Her feelings of shame do not abate, and she finds it increasingly uncomfortable to feel this way. She encounters a sign for a bar. Having previously learned that alcohol can reduce feelings of shame, Ada enters the bar and imbibes heavily (Figure 1c). Though drinking reduces her aversive internal state in the short term, Ada returns home drunk, leading to further conflict with Darius.

### I. DECIDING WHETHER TO ACT

When Ada left the apartment, she was acting to change her emotional state. We propose that Ada's perception of whether she could change her state guided this decision (Table 1.1). Crucially, although we frame this as a 'decision' about perceived controllability, such computations are not necessarily deliberative and can be rapid and implicit, especially for more engrained behavioral patterns (Huys & Dayan, 2009). Whereas appetitive contexts promote approach (Panksepp, 2004; Wasserman et al., 1974), aversive contexts promote multifarious behaviors (Blanchard et al., 2005; Bolles, 1970; McNaughton & Corr, 2004) including fighting, fleeing, freezing, and passive avoidance (McNaughton & Gray, 2000). In an aversive context, the choice to act depends on how the threat is appraised: perceiving it as distant and beyond behavioral control elicits passive avoidance, while

perceiving it as close and controllable (Boureau & Dayan, 2011) promotes active escape behaviors that exert control over the threat (e.g., fighting; Lloyd & Dayan, 2016; McNaughton & Gray, 2000).

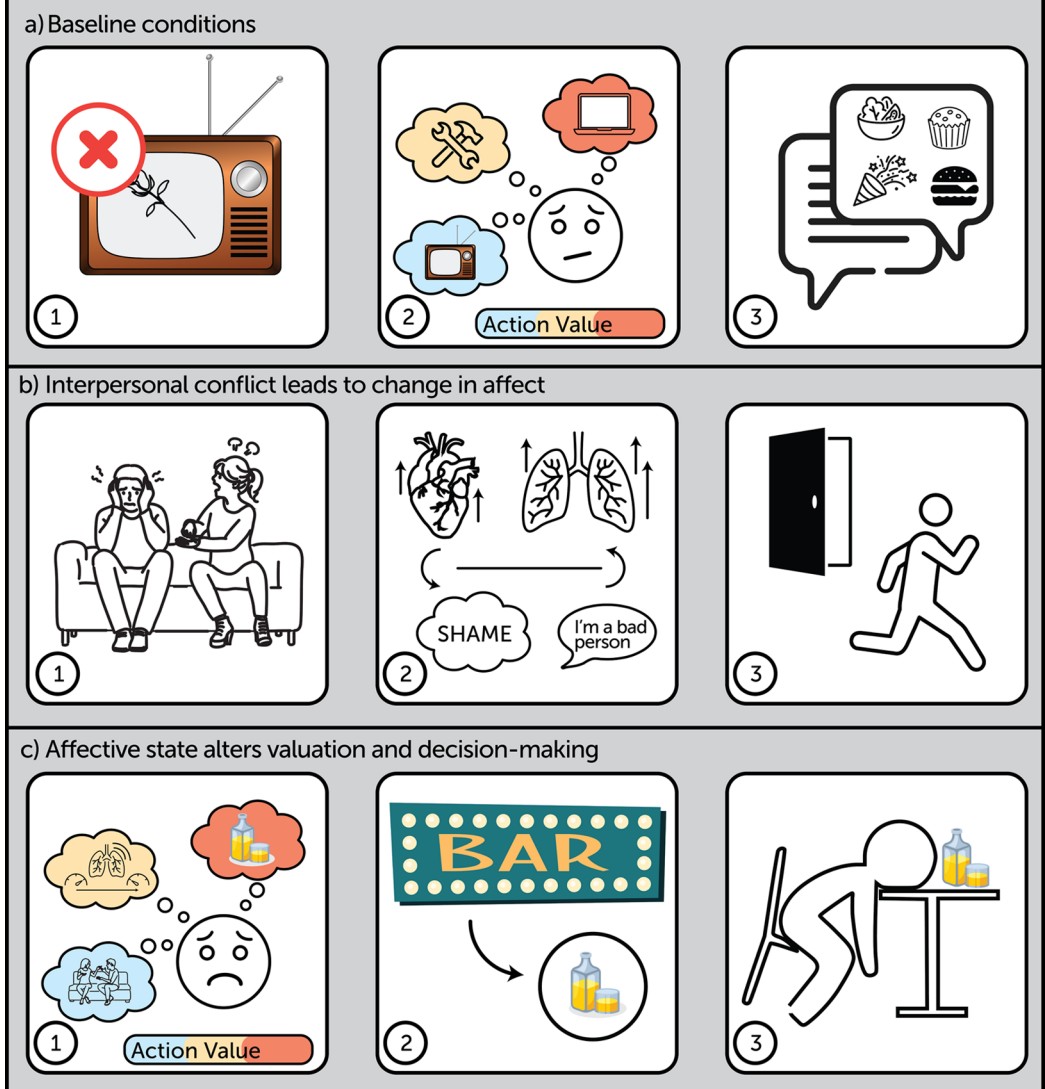

**Figure 1** Example of affect-based impulsivity. a) Baseline conditions. (i) Ada and Darius are watching a TV show but are experiencing technical difficulties. The picture resolution is poor due to the placement of the digital antenna. (ii) Ada considers different actions: watching the show on her laptop, fixing the TV, or continue watching the show on the TV (despite the poor resolution). (iii) Ada receives a text inviting her and Darius to a potluck the next evening. b) Interpersonal conflict leads to change in affective state. (i) Ada and Darius disagree over what baked good to bring to the potluck. (ii) Ada experiences an increase in negative affect and labels her emotional experience "shame." (iii) Too distressed to continue the conversation with Darius, Ada decides to leave the apartment. c) Negative affect alters valuation of different actions and motivates impulsive behavior. (i) Ada's intense emotional state persists, and she considers actions that she anticipates will reduce her current negative affective state: drinking alcohol, using a different emotion regulation skill (e.g., deep breathing), and returning home to talk through the conflict with Darius. (ii) While on her walk, Ada encounters a sign for a bar, which functions as a conditioned reward cue. (iii) Ada vigorously pursues the reward – alcohol – and quickly becomes drunk.

Deciding to act (i.e., active escape) versus not act (i.e., passive avoidance) recruits different learning systems. Passive avoidance largely depends on the Pavlovian system, which supports stimulus-outcome learning, such as which cues are associated with the threatening context (Sutton & Barto, 2018). By learning these associations, the organism can anticipate and avoid similar contexts in the future (Cartoni et al., 2016; Niv et al., 2006). In contrast, active escape principally depends on the instrumental system (Dorfman & Gershman, 2019; Moscarello & Hartley, 2017), which supports action-outcome learning (Sutton & Barto, 2018), including which behavior will eliminate a threat. From this vantage point, Ada's belief that she can alter her feelings of shame motivates her to act, and her behavior is primarily under instrumental control.

## II. NARROWING THE SET OF ACTIONS ACTIVELY CONSIDERED

Emotions function as a form of metareasoning, shaping the states and actions an organism considers (Anderson & Adolphs, 2014; Huys & Renz, 2017; Levenson, 2011; Panksepp, 2004; Sander et al., 2018) and promoting emotion-congruent responses (Anderson & Adolphs, 2014; Levenson, 2011; Panksepp, 2004). We propose that features of internal states act as conditioned cues, shifting an organism's goals and motivating actions to achieve these goals (Table 1.2; Cartoni et al., 2016). In our example, Ada's shift in internal state (neutral to shame) redirects her goals from planning to watch The Bachelor (Figure 1a-ii) to reducing intense aversive feelings. When a

walk fails to help, she considers additional emotion-congruent actions that align with her current goal – including (1) drinking alcohol, (2) using another emotion regulation skill, and (3) returning home to work through the argument with Darius (Figure 1c-i).

Schreiber and Hallquist **89**
*Computational Psychiatry*

| STAGE OF DECISION-MAKING | COGNITIVE AND REINFORCEMENT LEARNING MECHANISMS | NEUROCOMPUTATIONAL AND NEUROENDOCRINE TARGETS |
|---|---|---|
| i) Whether to act? | • Controllability determines whether to act<br>• When uncontrollable, Pavlovian learning predominates<br>• When controllable, action largely under instrumental control | • Computations of controllability encoded in PFC, as well as BNST, insula, and posterior cingulate gyrus<br>• When uncontrollable, serotonin inhibits behavior<br>• When controllable, dopamine (DA) motivates behavior |
| ii) Which actions to consider? | • Emotions alter goals and narrow action set to affect-congruent responses<br>• Pavlovian system supports learning conditioned cues of emotional state | • Salience network, primary and secondary somatosensory cortices, insula, hypothalamus are modulated by internal state<br>• Corticotropin releasing hormone initiates release of stress hormones<br>• Emotional state alters action set via related neural circuits (e.g., co-activation of circuits, hippocampal replay, PAG) |
| iii) How to decide among actions? | • Cached values of actions depend on prior learning when in similar affective state<br>• Actions are evaluated using less deliberative reasoning<br>• Model-free learning is frequently preferred, especially with constrained cognitive resources<br>• When model-based reasoning is used, computations simplified (e.g., shortening a simulation after a large loss) | • Glucocorticoids (GCs) and norepinephrine (NE) released as part of stress response<br>• High levels of NE and DA in PFC hamper effective communication between ensembles of neurons<br>• Less efficient processing in PFC leads to reduction in complex model-based computations |
| iv) How vigorously to engage in action? | • Heightened vigor for affect-congruent actions<br>• Appetitive Pavlovian-to-Instrumental Transfer (PIT) accounts for heightened vigor<br>• Appetitive PIT partly reflects opportunity cost associated with inaction | • Affect-related increases in GCs enhance reactivity of DA receptors in NAcc shell<br>• Heightened sensitization of mesolimbic DA reward circuit leads to enhanced pursuit of rewards associated with appetitive cues |

**Table 1** How does affective state shape decision processes across the four stages of a decision? *N.B.* Though our focus is on impulsive behaviors that are maladaptive, we anticipate that similar mechanisms may explain adaptive responses to an emotion (e.g., grizzly bear sighting → fear → freeze).

## III. EVALUATING THE ACTION SET

Once the action set has been winnowed, an organism must arbitrate among the available actions. In RL terms, these instrumental actions are either under control of the habitual or goal-directed system (Dolan & Dayan, 2013). Goal-directed decisions depend on evaluating the expected value obtained by taking certain actions (Niv et al., 2006). In a simple environment, an organism learns the value of different actions using straightforward model-free computations, tracking the expected return for each action based on historical outcomes.[1] However, in complex environments, like social interactions (FeldmanHall & Nassar, 2021), learning the value of different actions often relies on model-based algorithms (Dolan & Dayan, 2013) that are cognitively taxing (Otto,

---

1    Model-based and model-free learning are not categorically different, but are instead thought to lie on a continuum. These systems operate independently and in parallel, with computations from both systems guiding decision-making.

Gershman, et al., 2013). These algorithms require the organism to represent the environment's structure, the relationship among states, and the action-outcome associations in each state (Daw et al., 2005; Gläscher et al., 2010). To illustrate, if guided by model-free learning, Ada would choose an action by comparing the cached values of actions that are under consideration. Under model-based learning, she would rely on her mental model of Darius to predict his likely response to each alternative action.

So how do emotions alter these RL systems? Negative emotions are often intensely aversive and drive an organism to quickly arbitrate among actions (McNaughton & Corr, 2004; Mobbs, 2018). In our view, emotions simplify both model-free and model-based components of this arbitration process (Table 1.3). First, strong negative emotions can blunt computationally expensive model-based reasoning (Otto, Raio, et al., 2013; Schwabe et al., 2010; Schwabe & Wolf, 2009), prompting a shift toward model-free learning (Mkrtchian et al., 2017). Second, vis-à-vis state-dependent learning (Dickinson & Balleine, 1994; Mollenauer, 1971; Tovote et al., 2015), the cached values of available actions are re-mapped based on what was learned about those actions in similar states. Instrumental outcomes that were learned to be valuable when in a similar state acquire incentive value that is contingent on state (e.g., learning to use a vending machine when feeling thirsty; Dickinson & Balleine, 1994). When arbitrating among available actions, this adjustment to the cached value alters the value gradient among available actions (e.g., using the vending machine is a more highly valued action than it is typically).

Third, even when an organism engages in model-based reasoning to consider the consequences of different actions, emotions may shape this simulation process itself (as suggested in Huys & Renz, 2017). Emotions are associated with attentional biases (MacLeod et al., 1986; Mathews & MacLeod, 2005), and action sequences that are incongruent with the emotion may be selectively ignored when simulating paths forward (Huys & Renz, 2017). For example, when thinking through the consequences of a particular action sequence, organisms frequently fail to consider the long-run value if they encounter an (imagined) large negative outcome in the sequence (Huys et al., 2012; Lally et al., 2017). This tendency to "prune" action sequences that involve imagined negative outcomes is greater in people with elevated anxiety and depressive symptoms (Huys et al., 2012; Lally et al., 2017), and attentional biases toward threat are frequently found in anxiety and depressive disorders (Mathews & MacLeod, 2005). In our example, Ada could simulate what would happen if she attempted to work through the argument with Darius. She predicts that they will both raise their voices and then storm off to separate rooms (an aversive outcome). She may stop the simulation there and choose among available actions based on their immediate value, rather than simulating how they could eventually make amends.

## IV. ENGAGING IN A SELECTED ACTION: THE ROLE OF MOTIVATIONAL VIGOR

Emotions not only alter the perceived value of different actions but also enhance the vigor of a selected emotion-congruent action. We propose that appetitive Pavlovian-to-Instrumental Transfer (PIT; Table 1.4) is a key pathway through which negative affect enhances the pursuit of rewards (Cartoni et al., 2016). Emerging evidence indicates that this PIT-related invigoration results from the "opportunity cost" of not acting (Boureau & Dayan, 2011; Niv et al., 2006). That is, when an organism perceives that an aversive state can be improved (e.g., active escape), then the gap between the current and preferred states is wide, making each moment of inaction feel costly.

Suppose Ada has had a history of tumultuous relationships, and she thus experiences relationship discord as particularly distressing (via an aversive Pavlovian association). Consequently, the argument with Darius is especially distressing to her. Each moment of inaction feels costly, driving actions to reduce negative affect. When she decides to drink alcohol, she does so vigorously (McNamara et al., 2024). Had Ada's prior relationships been more stable, her distress in this situation might have been less acute. The perceived cost of inaction would have been lower, and her motivation to escape these feelings would be weaker.

# NEUROCOMPUTATIONAL ACCOUNT OF AFFECT-BASED IMPULSIVITY

Thus far, we have described a model of affect-based impulsivity in cognitive terms: 1) Deciding to act depends on perceptions of controllability; 2) The Pavlovian system sculpts which actions are considered, enhancing emotion-congruent actions; 3) Intense negative emotions blunt model-based reasoning, altering valuation of the action set; and 4) Appetitive PIT explains enhanced motivational vigor for the selected emotion-congruent action. We now turn to the neurobiological basis of these affect[2]-related shifts in decision-making. First, we propose that the decision to act depends on computations of controllability that are encoded in the prefrontal cortex (PFC). Second, which actions are considered depends on affective state. Finally, we describe the roles of dopamine (DA), glucocorticoids (GCs), and norepinephrine (NE) in, third, altering the valuation of available actions and, fourth, amplifying vigor for selected actions.

## I. THE BRAIN BASIS OF CONTROLLABILITY

Goal-directed decision-making relies on the expectation that an action will be instrumentally effective in achieving a desired outcome, such as alleviating a negative emotional state. Recent findings in computational cognitive neuroscience show that humans dynamically track the predictability (Dorfman & Gershman, 2019) and controllability (Ligneul et al., 2022) of an environment, and these computations occur in brain regions such as the PFC (Ligneul et al., 2022), bed nucleus of the stria terminalis (BNST), insula, and posterior cingulate gyrus (Limbachia et al., 2021). Critically, activity in these regions governs whether a stressor leads to action or inaction (i.e. "learned helplessness"; Amat et al., 2005; Table 1.1).

Building on these insights, we propose that computations of controllability are encoded in the PFC (Ligneul et al., 2022), as well as related brain regions (Limbachia et al., 2021), and gate the extent to which DA or serotonin predominates over behavior. Historically, DA has been thought to guide approach behavior in appetitive contexts and serotonin to inhibit punished behaviors[3] in aversive contexts (Boureau & Dayan, 2011). However, DA is involved in motivating behavior in both appetitive and aversive contexts (Boureau & Dayan, 2011; Guitart-Masip et al., 2014). In aversive contexts, by virtue of bringing the organism into a more desirable state, escape is encoded as a reward, and DA invigorates behaviors that help the organism escape (Lloyd & Dayan, 2016). Crucially, these DA-dependent behaviors only occur when the outcome is perceived as controllable (for review, see Boureau & Dayan, 2011). When behavior has no effect on the aversive environment, serotonin is released into mPFC and inhibits behavior (e.g., in learned helplessness experiments; Bland et al., 2003; Boureau & Dayan, 2011).

## II. AFFECTIVE BRAIN STATE SHAPES THE ACTION SET

Once an organism decides to act, its emotional state promotes affect-congruent actions – a process that depends on a corresponding brain state. Though there is debate about how to conceptualize and study emotions (Anderson & Adolphs, 2014; Barrett et al., 2007; Levenson, 2011; Panksepp, 2004; Sander et al., 2018), certain neural systems are nonetheless consistently implicated in emotional experiences (Table 1.2). Emotional experiences often begin when a motivationally relevant stimulus is detected, engaging orienting processes (Sander et al., 2018) supported by the ventral attention system (e.g., temporoparietal junction; Kincade et al., 2005) and the salience network (e.g., dorsal anterior cingulate cortex, anterior insula; Seeley et al., 2007). Then, there is a cascade of neuropeptides that modulate neural activity and organize a shift in internal state (Flavell et al., 2022). In certain instances, such as the presence of a threat, corticotropin-releasing hormone (Vale et al., 1981) is released and facilitates the stress response.

---

2    We use affect and emotion interchangeably when discussing the brain basis for how internal state alters decision processes, though we recognize that emotion theorists argue for a distinction between the two (Barrett et al., 2007).

3    Though serotonin is most consistently implicated in behavioral inhibition, this effect is not universal. For example, SSRIs can increase escape behaviors during a forced swim test in rats (Detke et al., 1995).

In parallel, brain systems regulating the autonomic nervous system (e.g., hypothalamus) modulate sympathetic and parasympathetic activity (e.g., increased heart rate). These changes are detected through interoception (primary and secondary somatosensory cortices) and are integrated into a multimodal representation of the body (insula; Critchley et al., 2004). As additional information about the stimulus is gathered through sustained attention, the estimate of the stimulus's relevance is adjusted, which may further modulate relevant circuits (Gross, 2015). Conceptual knowledge, prior experience, and language may also inform estimates of relevance (Barrett et al., 2007).

Once this affective brain state has been constructed, there are many pathways by which state shapes how the action set is narrowed (Table 1.2; Tovote et al., 2015). Some affect-congruent behaviors (e.g., freezing) are hard-wired, mediated by evolutionarily preserved midbrain structures (periaqueductal gray; Graybiel, 2008). Other pathways involve hippocampal replay, which helps retrieve actions that were effective when previously experiencing a similar emotional state (Carr et al., 2011). Co-activation (Tovote et al., 2015) of the same circuits that constitute an affective state (Barrett et al., 2007) may also shape the set of actions being considered (Huys & Renz, 2017).

## III. NEGATIVE AFFECT HAMPERS GOAL-DIRECTED COMPUTATIONS: THE ROLES OF GLUCOCORTICOIDS AND NOREPINEPHRINE

Once the action set has been narrowed, the organism selects among available actions using a combination of learning algorithms, which help the organism select the action associated with the highest expected value. Our model proposes that negative affective states lead the organism to rely primarily on model-free reasoning and to use shortcuts that simplify model-based computations. Building on the established links of stress and negative affect with GCs and NE (Blair et al., 2008; W. A. Brown & Heninger, 1975; Dickerson & Kemeny, 2004; J. R. Piazza et al., 2013), we consider how increases in GCs and NE may account for this shift in reasoning (Table 1.3).

Stress has widespread effects on cognition (Lupien et al., 2007; Starcke & Brand, 2012), including reduced cognitive performance for computations depending on mPFC functioning (Arnsten, 2009; de Berker et al., 2016; Schwabe & Wolf, 2013). These effects of stress are mediated by changes in intracellular signaling of PFC cells (Arnsten, 2009). NE enhances coherent firing of cells receiving similar information, thereby increasing the "signal" in brain regions performing a cognitive function. Conversely, DA in PFC decreases cell firing in response to motivationally irrelevant information, thus reducing the "noise" in surrounding brain regions. The interaction between NE and DA yields an inverted-U relationship between NE and DA on PFC functioning (Arnsten, 2009). At moderate levels of prefrontal NE and DA, neuronal ensembles can effectively communicate to perform complex computations. Yet, when NE and DA are very high, altered signaling in PFC interferes with effective communication among ensembles needed for complex computations. This observation aligns with the broader literature on stress-induced modulation of GCs and NE that blunt goal-directed decision-making and reduce model-based learning (Otto, Raio, et al., 2013; Schwabe et al., 2010, 2011, 2012). We suggest that affect-dependent reductions in model-based reasoning are mediated by these stress-related changes in PFC functioning.

## IV. AFFECT-RELATED CHANGES IN THE MESOLIMBIC DOPAMINE REWARD CIRCUIT ACCOUNT FOR INVIGORATED PURSUIT OF REWARD

In conjunction with these functional changes in PFC, we further propose that negative affect enhances Pavlovian influences on behavior – including invigorated pursuit of rewards – through its effects on the mesolimbic reward system (Table 1.4). To provide context, let us first consider how DA invigorates behavior.

**Enhanced pursuit of rewards: the role of dopamine.** Heightened mesolimbic DA reactivity to reward cues increases experiences of "wanting," or craving (Berridge & Robinson, 2016). "Wanting" of the cue, which is related to sign-tracking, typically involves increased behavioral engagement

with the reward cue, partly via Pavlovian mechanisms (Anselme et al., 2013; Morrison et al., 2015). Individual differences in sign-tracking predict greater PIT-dependent (Garofalo & di Pellegrino, 2015) approach toward and actions involving the reward cue itself (e.g., a rat licking the lever that predicts food). In the context of aversive states where the outcome is perceived as controllable, DA invigorates escape behaviors that promote safety and that reduce the aversiveness of the organism's internal state (Boureau & Dayan, 2011).

The nucleus accumbens (NAcc) is a central node of the mesolimbic reward circuit involved in learning from surprise: DA-related modulation of NAcc scales parametrically with the extent to which the outcome is better than expected (Glimcher, 2011). NAcc has two subregions, core and shell, which exhibit dissociable roles in learning. Whereas the core tracks the reward rate of the environment, enhances motivation, and invigorates behavior toward any reward cue (e.g., sign for bar increasing pursuit of *substances*), the shell motivates behavior toward rewards that are associated with specific cues (e.g., sign for bar increasing pursuit of alcohol; Corbit & Balleine, 2005; Floresco, 2015). Indeed, sign-tracking depends on DA reactivity in the NAcc shell (DiFeliceantonio & Berridge, 2012; Mahler & Berridge, 2012; Morrison et al., 2015; Warlow et al., 2017), consistent with its broader role in PIT (Corbit & Balleine, 2005).

**How does negative affect alter dopamine activity in reward circuits?** Stressors frequently elicit negative affect (Dickerson & Kemeny, 2004) and alter reactivity to reward cues. Stress sensitizes DA receptors in NAcc shell, boosting firing rate in response to reward cues and increasing pursuit of specific rewards (Marinelli & Piazza, 2002; Peciña et al., 2006; P. V. Piazza & Le Moal, 1996). The sensitization of mesolimbic reward circuitry is partly driven by glucocorticoids (GCs): high levels of circulating GCs activate NAcc shell glucocorticoid receptors (GRs; Marinelli & Piazza, 2002), enhancing DA-dependent activity and increasing the Pavlovian influence of reward cues (Marinelli & Piazza, 2002). Thus, during negative emotional states, concurrent increases in GCs enhance sensitivity to reward cues and motivate vigorous pursuit of these rewards.

## PATHS FORWARD

We have articulated a decision neuroscience account of affect-based impulsivity that provides specific testable hypotheses regarding neuroendocrine systems, neural substrates, and neurocomputational processes. As shown in Figure 2, negative affect leads to increases in GCs and NE, which in turn dampen goal-directed decision-making by impairing mPFC functioning (Arnsten, 2009; Schwabe et al., 2010, 2011, 2012; Schwabe & Wolf, 2013). Second, and in parallel, GCs sensitize the mesolimbic DA pathway (Marinelli & Piazza, 2002; P. V. Piazza & Le Moal, 1996; Rougé-Pont et al., 1993), thereby increasing the Pavlovian influence of reward cues on decision-making (Berridge & Robinson, 2016; Peciña et al., 2006) and invigorating pursuit of outcomes associated with these cues.

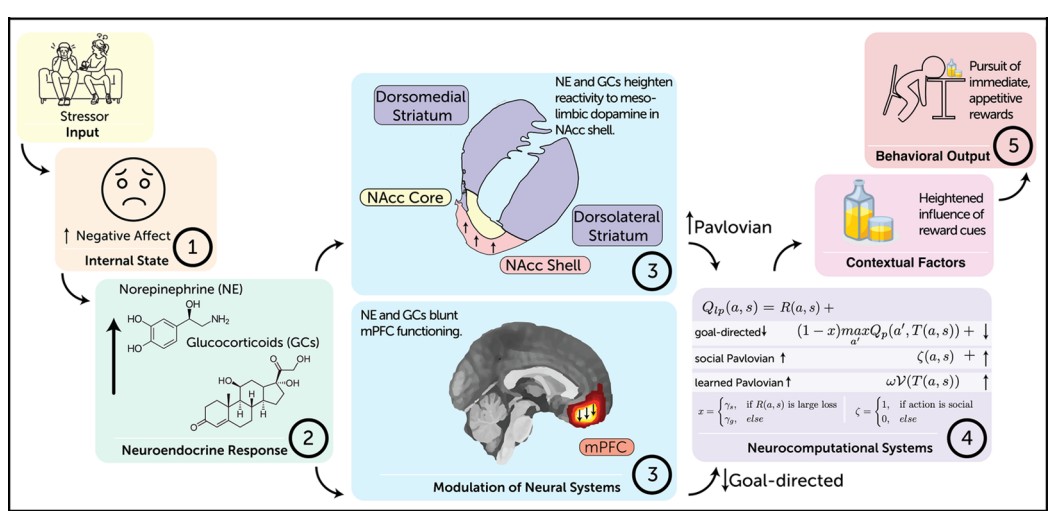

**Figure 2** A stressor (1) induces negative affect. Increase in negative affect elicits concomitant changes in circulating levels of (2) norepinephrine (NE) and glucocorticoids (GCs). (3) GCs enhance dopamine (DA) reactivity in nucleus accumbens (NAcc) shell, and NE and GCs blunt medial prefrontal cortex (mPFC) functioning. (4) Altered functioning in mPFC and NAcc shell results in altered balance of Pavlovian and goal-directed decision systems. Heightened influence of Pavlovian systems on decision-making amplifies influence of reward cues and (5) invigorates pursuit of rewards.

Schreiber and Hallquist **94**
*Computational Psychiatry*

Investigating these hypotheses requires advanced methods that can dissect a decision into latent components and corresponding neural circuits. Although many neuroscientific approaches may be used to study the neural bases of affect-based impulsivity, we consider decision neuroscience to be particularly promising. Decision neuroscience integrates Bayesian decision theory (Dayan & Daw, 2008; Huys, Daw, et al., 2015) with model-based cognitive neuroscience (Palmeri et al., 2017) to explore the neurocomputational mechanisms of behavioral phenomena (Dreher & Tremblay, 2016). This approach has advanced our understanding of the mechanisms involved in perceptual, value-based, and social decision-making (Dreher & Tremblay, 2016). Pharmacological and neuromodulatory methods promise to further strengthen the inferences of this research. Whereas traditional clinical neuroimaging methods are correlational (e.g., fMRI; Vytal & Hamann, 2010), transcranial direct-current stimulation (Stagg & Nitsche, 2011), transcranial magnetic stimulation (Wassermann et al., 2008), and pharmacological manipulations allow researchers to manipulate neural systems to test the causal role of a circuit or a brain region, potentially advancing mechanistic accounts of psychological phenomena (Allen et al., 2020).

Using these methods, our hypotheses could be tested effectively in four stages: First, test effects of a negative affect induction on learning and decision-making (Figure 2-1) in a task that can distinguish between Pavlovian and goal-directed decision systems (Figure 2-4; Gläscher et al., 2010; Huys et al., 2012). Second, link concomitant changes in GCs and NE to the balance of these decision systems (Figure 2-2). Third, use fMRI to identify neural correlates (Figure 2-3). Finally, employ a pharmacological manipulation of GCs and NE to characterize their role in shifting network dynamics and neural circuits. Together, these steps could reveal new insights into the neural bases of how negative affect alters decision-making, inform existing neurocognitive accounts of affect-based impulsivity, and yield findings with translational implications.

## LIMITATIONS AND FUTURE DIRECTIONS

Our account focuses on the within-person effect of negative affect on decision-making, but does not address *why* people vary in proneness to affect-based impulsivity (Berg et al., 2015; Cyders & Smith, 2008). We anticipate that our model could be expanded to address this open question. For instance, people vary in their appraisals of controllability, with low perceptions of control over threat linked to anxiety and depressive disorders (Cheng et al., 2013). How people tend to appraise controllability could impact whether they *act* to alter their emotion. Relatedly, people vary in the extent to which they tend to rely on model-based reasoning (Gillan et al., 2016; Patzelt et al., 2019), and trait affect-based impulsivity is associated with lower model-based reasoning – independent of a person's current emotional state (Patzelt et al., 2019). This deficit in model-based reasoning could compound the effect of negative emotions on action selection, perhaps via the mechanisms we proposed above. Third, our account emphasizes that affect-related increases in GCs lead to heightened DA reactivity in the mesolimbic reward circuit (Figure 2). There are between-person differences in GR sensitivity to GCs, reflecting effects of genetics and chronic stress (Kosten et al., 2002; Ortiz et al., 1996; Rougé-Pont et al., 1993), and this sensitivity impacts the potency of GCs on DA reactivity (P. V. Piazza & Le Moal, 1996). A natural next step in developing our account would be to examine whether GR sensitivity is related to trait affect-based impulsivity.

It is also worth highlighting constructs and phenomena that are not addressed in the present account of affect-based impulsivity. First, our proposal focuses on how negative affect – not positive affect – enhances impulsive behaviors. Second, our proposal does not directly consider explicit emotion regulation strategies that people commonly deploy, including suppression and reappraisal (Gross, 2015). Indeed, enacting effective emotion regulation strategies may fall within the set of options a person evaluates when considering how to respond to their emotional state. Third, negative affect may increase habitual behavior (Schwabe et al., 2010, 2012; Schwabe & Wolf, 2009, 2013), yet our model does not directly address the role of habit. Notably, habitual control relies on model-free learning systems (for review, see Dolan & Dayan, 2013). Thus, it may be fruitful to extend our account to consider habitual impulsive behaviors. Such an extension would require further consideration of how repeated experiences consolidate model-free representations into inflexible stimulus-response policies, and whether this process is dependent on affect-based

impulsivity. Fourth, certain emotions affect the decisiveness with which a person takes action (e.g., anger promotes decisiveness; Lerner & Tiedens, 2006). Altered decisiveness may be related to lower-level cognitive processes like decision threshold in drift diffusion models (Ratcliff & McKoon, 2008). Fifth, we primarily focus on the role of GCs and NE, yet other neuroendocrine systems (e.g., oxytocin) interact with GCs and NE, affect DA reactivity, and may separately alter the same decision systems through other pathways (Crockett & Fehr, 2013; Huys et al., 2012). Finally, even as we have described how emotions impact distinct stages of decision-making, which are relevant to impulsive behaviors, not all actions that are a response to an emotion lie within the scope of the model. For example, some affect-congruent responses are reflexive or hard-wired (e.g., freezing). As these behaviors are not learned, our hypotheses on arbitration are irrelevant. Similarly, the ways in which emotions alter simulations in model-based control is only relevant in environments that are complex enough to necessitate model-based reasoning (though it's worth noting that model-based computations are evident even in circumstances previously assumed to not necessitate model-based reasoning; Collins & Frank, 2012).

There are several methodological limitations that we must also acknowledge. Extant research on the neurocognitive substrates of trait affect-based impulsivity has been conducted in humans, and we must contend with the limitations present within the field of human neuroscience, including high type-I error rate (Szucs & Ioannidis, 2020), small clinical samples (Marek et al., 2020; Poldrack et al., 2017; Szucs & Ioannidis, 2020), different preprocessing and analysis methods (Collaboration, 2015; Esteban et al., 2019; Ioannidis, 2005; Power et al., 2012), and often poor reliability of behavioral tasks (Hedge et al., 2018). Use of larger samples and of state-of-the-art acquisition, preprocessing (Esteban et al., 2019), and analysis (V. M. Brown et al., 2020; Haines et al., 2020; Price et al., 2019; Rouder & Haaf, 2019) methods would help address these concerns. Finally, additional challenges arise when employing a decision neuroscience approach. Researchers must specify, estimate, and test plausible computational models of behavior (Daunizeau et al., 2014; Kruschke, 2014).

## CONCLUSION

We propose a model of affect-based impulsivity that explains behavior during negative affective states in terms of Pavlovian and goal-directed decision systems. Our model proposes that affect-related increases in glucocorticoids (GCs) and norepinephrine (NE) shift the balance of these decision systems in two key ways: (1) they blunt mPFC functioning, reducing goal-directed decision-making, and (2) they sensitize the mesolimbic DA pathway, enhancing vigorous pursuit of rewards. Decision neuroscience methods provide a framework for testing this account (Gureckis & Love, 2015; Love, 2015). Pharmacological and neuromodulatory methods are well suited for interrogating related neurocomputational systems (Allen et al., 2020). Altogether, a decision neuroscience account of affect-based impulsivity can advance our understanding of this transdiagnostic construct and yield new treatment targets for several psychiatric disorders.

## ACKNOWLEDGEMENTS

We would like to thank Eric A. Youngstrom, Stacey B. Daughters, Timothy A. Allen, and Alexandre Y. Dombrovski for offering helpful feedback on earlier versions of this manuscript. This manuscript is published as a preprint on PsyArXiv: https://osf.io/preprints/psyarxiv/jvmzs.

## FUNDING INFORMATION

This work was supported by the National Institute of Mental Health [T32 - MH019986 (AMS); R01—MH119399 (MNH); R01 – MH048463 (MNH)].

## COMPETING INTERESTS

The authors have no competing interests to declare.

## AUTHOR CONTRIBUTIONS

AMS – conceptualization, visualization, original draft, and editing; MNH – review and editing.

## AUTHOR AFFILIATIONS

**Alison M. Schreiber** orcid.org/0000-0003-3681-4252
Department of Psychiatry, University of Kentucky, Lexington, USA; Department of Psychology and Neuroscience, University of North Carolina at Chapel Hill, Chapel Hill, USA
**Michael N. Hallquist** orcid.org/0000-0001-5894-8038
Department of Psychology and Neuroscience, University of North Carolina at Chapel Hill, Chapel Hill, USA

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

Schreiber and Hallquist    **103**
*Computational Psychiatry*

**TO CITE THIS ARTICLE:**

**Submitted:** 15 September 2025

