## [Reviewer Report · Peer Review History Round 1, Silvia Lopez-Guzman]

## submission-comments

In this review article, the authors propose an overarching theory of impulsive behavior in response to negative affect through the lens of decision neuroscience. There is much to like about this paper. It is well written, it clearly presents its ideas and provides a bird’s eye view of how neural systems could be contributing to supporting these processes, which I think would be useful to a wide audience. While I find this narrative review compelling and worth publishing, I do have a few questions about the logic and the claims. I think addressing these could be helpful in improving its impact.

Major points:

I find the core of this conceptualization compelling. That is, that emotional state shapes arbitration between model-free and model-based processing. In the manuscript this component corresponds to section III. The authors explain three mechanisms that possibly explain a shift from model-based to model-free decision-making. First, emotion-driven restriction of computationally-expensive processes; second, re-mapping of cached values for available actions based on previous learning in the same emotional state; and third, negative emotions can themselves tint the possible outcomes and values simulated in a model-based way. In my opinion, these three mechanisms should be better explained, expounded on, and substantiated with available evidence. Particularly, the second mechanism seems very superficially described. I am not sure what is specifically meant by “re-mapped” cached action values. Perhaps, providing concrete examples of this and even a descriptive figure would be ideal to clarify these ideas.

As I mentioned, section III on action set evaluation seems particularly sound to me. However, I am less convinced about sections I and II are necessary processes. That is, if reacting to negative affect is already a learned response, the agent need not “decide whether to act”. Similarly, the learned reactive behavior may be so automatic that the agent may not even consider alternative responses, thereby bypassing the “narrowing action set” step. Can the authors clarify the importance of these steps in their general conceptual model or alternatively, explain that these steps are relevant for the specific example used to illustrate it? Could it be that these processes are relevant before behavioral patterns are learned and become the default?

Section I on deciding whether to act makes this process sound deliberative. The agent has some belief over controllability and selects a type of behavior on the basis of this evaluated belief. Is there evidence that this is conscious or intentional? It is also possible that the selection of different models of behavioral control (passive versus active) could be associated with the presence of other existing comorbidities, for example, anxiety, which this section does not discuss.

Emotion regulation processes like reappraisal and suppression are not really part of the neurocomputational process proposed here but are undoubtedly important (and even present in the example story of Ada, i.e. taking a walk, working things out with Darious, etc). I think the authors should acknowledge that this is a missing piece or explain why it is beyond the purpose of the review.

Minor points

I suggest expanding the references cited the section on the neurocomputational bases of controllability to include Bland et al., 2003 https://www.nature.com/articles/1300206, and to mention that the circuit extends beyond prefrontal cortex (Limbachia et al., 2021 https://www.nature.com/articles/s42003-020-01537-5?fromPaywallRec=true).

What exactly is meant by impulsivity in this paper? A rash decision? A model-free decision? A risky decision? A suboptimal decision? A shift in delay discounting? I think it would be good to define impulsive behavior a bit better here.

Typo on page 13: “These effects of stress are (be) mediated by changes…”

## peer-review-recommendation

Revisions Required

---

## [Reviewer Report · Peer Review History Round 1, Henry Chase]

## submission-comments

Schreiber and Hallquist review literature on computational mechanisms underlying affect based impulsivity. Briefly, the review seems to revolve around the same conundrum as that outlined by Huys and Renz, namely that pruning the model-based system provides an elegant account of the impact of emotion on decision making but that it is somewhat underdetermined, and valuation changes could have similar impact. Overall, the review is difficult to read, achieving baffling levels of self-contradiction. It is true that there are some interesting review articles written at the interface of phenomenological and behavioral (e.g. decision making paradigms) research which can reach across traditional research ‘silos’ and inspire fresh thinking. Whether or not such laudable aims drove the present work, perhaps some reconsideration of the authors’ objectives might be worthwhile.

Stylistically, the authors allow themselves to cite selectively - key empirical work on impulsivity and model-based decision making is not cited (e.g. Gillan/Daw elife, Patzelt/Gershman Biological Psychiatry) - and yet rule out a priori discussion of the strongest body of empirical work (response inhibition), on the grounds that it traditionally not analyzed using generative models. If generative models are so important, why not cite key work on impulsivity which has actually used them? If generative models are not important, why not cite the strongest body of empirical work? And then, what if generative models (e.g. accumulator models) were to be used to analyze response inhibition data? Are these data allowed back ‘in the fold’?

At some point the authors state that they are most interested in within-subject changes in emotion. But between subject difference are still apparent, at least implicitly, to derive a contrast. If the Pavlovian system is ‘more’ engaged, it is more engaged relative to what? A past version of the individual, or another person? Balanced discussion of within and between participant effects is generally typical throughout psychiatry, and individual differences are at the heart of psychiatry - this is a psychiatry journal.

On page 4 the modelling approach is described as ‘new’, but it draws heavily on existing accounts, many of which have been well rehearsed in the literature. The models in question are explanatorily powerful, meaning they can be applied widely as other authors have found.

On page 5, the authors ‘appreciate’ the value of cognitive neuroscience, and thankfully proceed to cite numerous such studies. Animal and computational models have great value in building accounts of impulsivity, but ideally we would want human cognitive neuroscience also to have a central role within a translational research program of impulsivity.

It isn’t counter-intuitive that aversive states should promote pursuit of appetitive cues - mood repair is obvious and well established within the addiction literature (e.g. by Koob and others).

Page 12 - serotonin doesn’t inhibit behavior, at least not universally. SSRIs increase escape behavior on a forced swim test. A citation is needed for DA increasing approach behavior when the outcome is perceived as controllable - it is true that DA does interact with contingency but approach behavior could simply mean approaching Pavlovian cues.

On page 19, the authors claim that their model does not address the role of habit, yet discuss cached values and a model-free system. It is possible to differentiate model free and habitual systems, but ultimately these terms arise from different paradigms but refer to something broadly similar. Overall, the authors should choose whether they are committing to a rather specific language in which e.g. model-free learning and habit can been distinguished, or a rather loose language in which ‘Pavlovian learning’ is unitary.

The review doesn’t appear to distinguish between impulsive action or choice, and within choice, risk taking and delay discounting. Again, this may be as a result of a desire for internal consistency with the cost of generalizability/scope.

## peer-review-recommendation

Resubmit for Review

---

## [Reviewer Report · Peer Review History Round 2, Silvia Lopez-Guzman]

## submission-comments

The authors have addressed all my suggestions satisfactorily.

## peer-review-recommendation

Accept Submission

---

## [Reviewer Report · Peer Review History Round 2, Henry Chase]

## submission-comments

The authors have made some improvement to the work. While these ideas, and similar ones, are popular in the field, I struggle with the fact that there seems to be several distinct computational routes to achieve the same outcome in this account. Emotions could bias decision making via a conditioned stimulus-response pathway; they could prune the potential model-based action repertoire to be considered; they could change the valuation of model-based actions; they could bias towards model free over model based control. Maybe this ambiguity serves as a form of encouragement for experimentalists to pin these ideas down, but usually you would look to theory to help with this rather than the other way around.

There is work by Brown/Price and others showing improved psychometrics with computational methods (already cited), but overall there is no guarantee that computational modelling offers any psychometric advantage. We would expect a psychometric improvement, of course, but in practice this is not necessarily the case. For example, Collins et al have had difficulty demonstrating generalizability across different assessments of learning rate parameters. I suppose one could say that you could keep fitting different models until psychometric improvement was observed, given that if structured variance that would drive test-retest reliability is ignored by one set of models, other models could be found which capture this variation. But it could be A) modest or B) irrelevant to the construct of interest. At bottom, then, psychometrics seems to be a non-sequitur and I’m concerned that any claim of a priori improvement is potentially misleading. It’s fairly obvious to the reader that the authors didn’t want to talk about response inhibition because it was outside of their focus on decision making rather than any psychometric issue.

The within/between person issue still stands - if a claim is made, i.e. Ada drinks vigorously, this implies a high intake on a relative or an absolute scale. How could such an absolute scale be defined? Generally we think of alcohol consumption within (human) culturally specified terms - hence, a relative scale - so, between subjects. It’s possible to think about purely in terms of pharmacological indices, but how could we define what ‘vigorous’ is in these terms? I don’t see how a purely within-subject account can be achieved in this context. I suppose an analogy would be sporting achievement - one could say that X improved their running time by some amount, but it would be natural to understand that achievement by reference to others. It might be possible to build a bioengineering account of human capacities, and show that this time is close to what can be achieved given biological constraints. But in this manuscript, almost all of the background work, citations etc, is all from a between-subject point of view, and the kind of framework that would be needed to establish within-subject assessment in absolute terms (i.e. independent of norming to a population) is not introduced. Concretely - let’s say I don’t drink anything all year, but then have a few glasses of wine at Christmas - in percentage terms, my intake and blood alcohol have gone up enormously (potentially infinite if it is normally zero). But I would guess this isn’t what the authors are getting at.

## peer-review-recommendation

Revisions Required

---

## [Reviewer Report · Round 1, Author Response]

Reviewer 1:

## Reviewer E

In this review article, the authors propose an overarching theory of impulsive behavior in response to negative affect through the lens of decision neuroscience. There is much to like about this paper. It is well written, it clearly presents its ideas and provides a bird’s eye view of how neural systems could be contributing to supporting these processes, which I think would be useful to a wide audience. While I find this narrative review compelling and worth publishing, I do have a few questions about the logic and the claims. I think addressing these could be helpful in improving its impact.

## Author’s response

Thank you for your thoughtful comments. We appreciate the critical points you raise below, as well as the positive comments about the strengths of the paper you highlight above. Altogether, we believe your comments have substantively strengthened our paper, and we are hopeful that we have sufficiently addressed your concerns.

## Reviewer E

Major points:

1. I find the core of this conceptualization compelling. That is, that emotional state shapes arbitration between model-free and model-based processing. In the manuscript this component corresponds to section III. The authors explain three mechanisms that possibly explain a shift from model-based to model-free decision-making. First, emotion-driven restriction of computationally-expensive processes; second, re-mapping of cached values for available actions based on previous learning in the same emotional state; and third, negative emotions can themselves tint the possible outcomes and values simulated in a model-based way. In my opinion, these three mechanisms should be better explained, expounded on, and substantiated with available evidence. Particularly, the second mechanism seems very superficially described. I am not sure what is specifically meant by “re-mapped” cached action values. Perhaps, providing concrete examples of this and even a descriptive figure would be ideal to clarify these ideas.

## Author’s response

We are heartened to hear that you find core features of our model compelling, and we appreciate the importance of fully describing each pathway by which emotions shape the arbitration of actions. In the revised text, we have significantly expanded on this section (pp. 10-11):

First, strong negative emotions can blunt computationally expensive model-based reasoning (Otto, Raio, et al., 2013; Schwabe et al., 2010; Schwabe & Wolf, 2009), prompting a shift toward model-free learning (Mkrtchian et al., 2017). Second, vis-à-vis state-dependent learning (Dickinson & Balleine, 1994; Mollenauer, 1971; Tovote et al., 2015), the cached values of available actions are re-mapped based on what was learned about those actions in similar states. Instrumental outcomes that were learned to be valuable when in a similar state acquire incentive value that is contingent on state (e.g., learning to use a vending machine when feeling thirsty; Dickinson & Balleine, 1994). When arbitrating among available actions, this adjustment to the cached value alters the value gradient among available actions (e.g., using the vending machine is a more highly valued action than it is typically).

Third, even when an organism engages in model-based reasoning to consider the consequences of different actions, emotions may shape this simulation process itself (as suggested in Huys & Renz, 2017). Emotions are associated with attentional biases (MacLeod et al., 1986; Mathews & MacLeod, 2005), and action sequences that are incongruent with the emotion may be selectively ignored when simulating paths forward (Huys & Renz, 2017). For example, when thinking through the consequences of a particular action sequence, organisms frequently fail to consider the long-run value if they encounter an (imagined) large negative outcome in the sequence (Huys et al., 2012; Lally et al., 2017). This tendency to “prune” action sequences that involve imagined negative outcomes is greater in people with elevated anxiety and depressive symptoms (Huys et al., 2012; Lally et al., 2017), and attentional biases toward threat are frequently found in anxiety and depressive disorders (Mathews & MacLeod, 2005).

## Reviewer E

2. As I mentioned, section III on action set evaluation seems particularly sound to me. However, I am less convinced about sections I and II are necessary processes. That is, if reacting to negative affect is already a learned response, the agent need not “decide whether to act”. Similarly, the learned reactive behavior may be so automatic that the agent may not even consider alternative responses, thereby bypassing the “narrowing action set” step. Can the authors clarify the importance of these steps in their general conceptual model or alternatively, explain that these steps are relevant for the specific example used to illustrate it? Could it be that these processes are relevant before behavioral patterns are learned and become the default?

## Author’s response

We agree that certain components of our model are more relevant when a behavioral pattern is emerging and has not yet become fully engrained. For example, model-based learning is more common early in learning. Consequently, the effects of emotions on the simulation process may be more pronounced early in learning.

We nonetheless contend that these steps can unfold very quickly and do not necessarily require deliberative reasoning or conscious thought. In our view, it is likely the case that each stage of the decision process is relevant for most instrumental actions that occur in response to a negative emotion. For example, if a person sees a grizzly bear 100 yards away, they will quickly appraise whether that bear is close enough to be within behavior control. Conversely, if the bear is quite close, then they must act swiftly to save their life. In a state of fear, their goal shifts (to staying alive). Fear-congruent responses (flight, flee, freezing) predominate the action set, and they evaluate which action to select using simplified computations. Whichever action they chose, it is likely they do so vigorously (given the urgency of the situation). Thus, even in this scenario where a person must respond very quickly, each stage of the decision process still unfolds. We now include text to more fully explicate ours views on this topic:

Although we describe these as sequential yet overlapping computations, we do not mean to imply that each stage involves conscious deliberation. Rather, we view these four aspects as interacting processes that may unfold rapidly and in parallel. pg. 6

Crucially, although we frame this as a ‘decision’ about perceived controllability, such computations are not necessarily deliberative and can be rapid and implicit, especially for more engrained behavioral patterns (Huys & Dayan, 2009). pg. 7

We note one important exception. Some affect-congruent responses are hardwired (e.g., freezing) or are reflexive, relying on evolutionarily preserved midbrain structures (e.g., PAG). Reflexes do not depend on evaluation of values. Thus, the action valuation and selection components of our model do not apply to reflexes.

To your broader point, it is true that some sub-components of our model may not always be relevant, even if each stage of the decision is still relevant. For example, we have hypothesized that emotions shape the simulation process during model-based planning. We predict that this will only occur in instances where the environment is complex enough to necessitate model-based control. Should the environment be simple, with model-free computations sufficing, such computations may be unnecessary. We now directly address this issue as a caveat under Limitations and Future Directions (pg. 22):

Finally, even as we have described how emotions impact distinct stages of decisionmaking, not all actions that are a response to an emotion lie within the scope of the model. For example, some affect-congruent responses are reflexive or hard-wired (e.g., freezing). As these behaviors are not learned, our hypotheses on arbitration are irrelevant. Similarly, the ways in which emotions alter simulations in model-based control is only relevant in environments that are complex enough to necessitate model-based reasoning (though it’s worth noting that model-based computations are evident even in circumstances previously assumed to not necessitate model-based reasoning; Collins & Frank, 2012).

## Reviewer E

3. Section I on deciding whether to act makes this process sound deliberative. The agent has some belief over controllability and selects a type of behavior on the basis of this evaluated belief. Is there evidence that this is conscious or intentional? It is also possible that the selection of different models of behavioral control (passive versus active) could be associated with the presence of other existing comorbidities, for example, anxiety, which this section does not discuss.

## Author’s response

We agree that beliefs around controllability can be conscious, slow, and deliberative (e.g., a depressed person thinking “nothing I do ever matters”), as well as fast and seemingly automatic (e.g., once you see a grizzly bear one hundred yards away, you know there’s nothing you can do to exert control over the bear). In our view, both cases still require an appraisal about the stressor’s controllability, even if that computation occurs quickly or defaults to a person’s typical perception of controllability (i.e., a prior). We now explicitly address this consideration on pg. 7 of the manuscript:

Crucially, although we frame this as a ‘decision’ about perceived controllability, such computations are not necessarily deliberative and can be rapid and implicit, especially for more engrained behavioral patterns (Huys & Dayan, 2009).

We appreciate that psychiatric comorbidities will impact people’s priors on controllability. We now directly address this issue on pg. 20 of the manuscript:

For instance, people vary in their appraisals of controllability, with low perceptions of control over threat linked to anxiety and depressive disorders (Cheng et al., 2013). How people tend to appraise controllability could impact whether they act to alter their emotion.

## Reviewer E

4. Emotion regulation processes like reappraisal and suppression are not really part of the neurocomputational process proposed here but are undoubtedly important (and even present in the example story of Ada, i.e. taking a walk, working things out with Darious, etc). I think the authors should acknowledge that this is a missing piece or explain why it is beyond the purpose of the review.

## Author’s response

We agree that emotion regulation processes are relevant to affect-based impulsivity and even referenced in our example. Even as these processes are highly relevant to affectbased impulsivity, we still view them as outside the explanatory scope of the review and proposed model. We articulate our reasoning on pg. 21:

Second, our proposal does not directly consider explicit emotion regulation strategies that people commonly deploy, including suppression and reappraisal (Gross, 2015). Indeed, enacting effective emotion regulation strategies may fall within the set of options a person evaluates when considering how to respond to their emotional state.

We also note that we indicate the potential use of our model for describing adaptive responses to emotions on pg. 49 of the manuscript (Table 1):

N.B. Though our focus is on impulsive behaviors that are maladaptive, we anticipate that similar mechanisms may explain adaptive responses to an emotion (e.g., grizzly bear sighting → fear → freeze).

## Reviewer E

Minor points

1. I suggest expanding the references cited the section on the neurocomputational bases of controllability to include Bland et al., 2003 https://www.nature.com/articles/1300206, and to mention that the circuit extends beyond prefrontal cortex (Limbachia et al., 2021 https://www.nature.com/articles/s42003-020-01537-5?fromPaywallRec=true).

## Author’s response

We appreciate these excellent reference suggestions. We recognize that controllability computations are not solely encoded in mPFC and have amended this section to highlight additional regions that are modulated by controllability. We also appreciate the suggestion to include the Bland article, which provides a mechanistic account for how controllability alters levels of DA and serotonin in mPFC. Below is a revised version of this section (pp. 13-14):

Recent findings in computational cognitive neuroscience show that humans dynamically track the predictability (Dorfman & Gershman, 2019) and controllability (Ligneul et al., 2022) of an environment, and these computations occur in brain regions such as the PFC (Ligneul et al., 2022), bed nucleus of the stria terminalis (BNST), insula, and posterior cingulate gyrus (Limbachia et al., 2021). Critically, activity in these regions governs whether a stressor leads to action or inaction (i.e. “learned helplessness”; Amat et al., 2005; Table 1.1).

Building on these insights, we propose that computations of controllability are encoded in the PFC (Ligneul et al., 2022), as well as related brain regions (Limbachia et al., 2021), and gate the extent to which DA or serotonin predominates over behavior. Historically, DA has been thought to guide approach behavior in appetitive contexts and serotonin to inhibit punished behaviors in aversive contexts (Boureau & Dayan, 2011). However, DA is involved in motivating behavior in both appetitive and aversive contexts (Boureau & Dayan, 2011; Guitart-Masip et al., 2014). In aversive contexts, by virtue of bringing the organism into a more desirable state, escape is encoded as a reward, and DA invigorates behaviors that help the organism escape (Lloyd & Dayan, 2016). Crucially, these DA-dependent behaviors only occur when the outcome is perceived as controllable (for review, see Boureau & Dayan, 2011). When behavior has no effect on the aversive environment, serotonin is released into mPFC and inhibits behavior (e.g., in learned helplessness experiments; Bland et al., 2003; Boureau & Dayan, 2011).

## Reviewer E

2. What exactly is meant by impulsivity in this paper? A rash decision? A model-free decision? A risky decision? A suboptimal decision? A shift in delay discounting? I think it would be good to define impulsive behavior a bit better here.

## Author’s response

It is certainly true that impulsivity refers to many behavioral tendencies, including heightened delay discounting, risky preferences, suboptimal decisions, and rash actions. We agree that is important to more clearly delineate what impulsivity means in the context of our manuscript. As such, we have added the following text to the introduction (pg. 5):

Second, impulsivity is not a unitary construct, referring to distinct behavioral tendencies ranging from heightened delay discounting to inhibitory control deficits to impulsive decision-making style (Caswell et al., 2015; Sharma et al., 2014). In our model, we focus on impulsive behaviors that are short-sighted and rash, reflecting a preference for immediate rewards, consistent with original psychological theories on trait affect-based impulsivity (Cyders & Smith, 2008; Smith et al., 2007).

## Reviewer E

3. Typo on page 13: “These effects of stress are (be) mediated by changes…”

## Author’s response

Thank you for bringing our attention to this issue. We have corrected the grammar of this sentence in the revised manuscript.

Henry Chase

## Reviewer H

Schreiber and Hallquist review literature on computational mechanisms underlying affect based impulsivity. Briefly, the review seems to revolve around the same conundrum as that outlined by Huys and Renz, namely that pruning the model-based system provides an elegant account of the impact of emotion on decision making but that it is somewhat underdetermined, and valuation changes could have similar impact. Overall, the review is difficult to read, achieving baffling levels of self-contradiction. It is true that there are some interesting review articles written at the interface of phenomenological and behavioral (e.g. decision making paradigms) research which can reach across traditional research ‘silos’ and inspire fresh thinking. Whether or not such laudable aims drove the present work, perhaps some reconsideration of the authors’ objectives might be worthwhile.

## Author’s response

You are correct that our goal in writing this article was to synthesize literature across diverse research traditions, with the goal of inspiring fresh thinking for how to study affect-based impulsivity. As you imply, cross-talk between research ‘silos’ is often limited, stymying progress in developing a mechanistic understanding for how affectbased impulsivity arises. We have made significant revisions throughout the manuscript (summarized above and below), which we hope more clearly communicates the aims of this manuscript, as well as delineating the scope of the narrative review.

## Reviewer H

• Stylistically, the authors allow themselves to cite selectively - key empirical work on impulsivity and model-based decision making is not cited (e.g. Gillan/Daw elife, Patzelt/Gershman Biological Psychiatry) - and yet rule out a priori discussion of the strongest body of empirical work (response inhibition), on the grounds that it traditionally not analyzed using generative models. If generative models are so important, why not cite key work on impulsivity which has actually used them? If generative models are not important, why not cite the strongest body of empirical work? And then, what if generative models (e.g. accumulator models) were to be used to analyze response inhibition data? Are these data allowed back ‘in the fold’?

## Author’s response

We appreciate that our original manuscript did not include seminal papers on model-based control, which are relevant to the paper. We have thus expanded the reference list to include Gillan et al. (2016) and Patzelt et al. (2019) on pg. 20.

We also appreciate the sizeable literature linking response inhibition deficits to trait affect-based impulsivity. We note that this literature is cited on pg. 4 of the manuscript. As you keenly point out, much of this work relies on summary statistics of task behavior, which have significant psychometric limitations – especially when the goal is to link summary indices to traits (Hedge et al. 2018). Indeed, a growing body of work shows that a domain-general bias toward inefficient evidence accumulation is associated with psychopathology (Sripada & Weigard, 2021), including impulsivity (Hall et al., 2021; Schreiber et al., 2025), and this bias can account for poor performance on inhibitory control tasks, as well as other neurocognitive tasks (Weigard et al., 2021). In light of this literature, inefficient evidence accumulation could account for many of the observed findings that link trait affect-based impulsivity with response inhibition. Given our focus on *underlying* computational processes that generate impulsive behavior, we do not believe citing this literature more thoroughly is appropriate. Nonetheless, we agree with your suggestion that generative models can be applied to response inhibition paradigms and may yield important insights that are relevant for affect-based impulsivity. We note that we mention this possibility as a future direction on pg. 21 of the manuscript:

Fourth, certain emotions affect the decisiveness with which a person takes action (e.g., anger promotes decisiveness; Lerner & Tiedens, 2006). Altered decisiveness may be related to lower-level cognitive processes like decision threshold in drift diffusion models (Ratcliff & McKoon, 2008).

We appreciate that we did not clearly state our concerns with the extant literature and have thus expanded on this issue in the introduction:

Studies to date have primarily relied on summary indices of emotion processing or response inhibition (e.g., rate of inhibitory failures), which only provide indirect evidence about lower-level cognitive processes (Gureckis & Love, 2015; Love, 2015) and often have poor psychometric properties (Hedge et al., 2018).

## Reviewer H

• At some point the authors state that they are most interested in within-subject changes in emotion. But between subject difference are still apparent, at least implicitly, to derive a contrast. If the Pavlovian system is ‘more’ engaged, it is more engaged relative to what? A past version of the individual, or another person? Balanced discussion of within and between participant effects is generally typical throughout psychiatry, and individual differences are at the heart of psychiatry - this is a psychiatry journal.

## Author’s response

We certainly agree that both within- and between-person effects are relevant to affect-based impulsivity. Our model is focused on within-person effects, with the hope that the model can be extended to capture between-person differences. Though we view this as a critical next step, there is not sufficient space in the current manuscript to consider all potential pathways that would lead someone to be prone to affect-based impulsivity. Nonetheless, we have extended our discussion of how our model might be extended to account for between-person differences under Limitations and Future Directions. Below is a revised version of this section:

We anticipate that our model could be expanded to address this open question. For instance, people vary in their appraisals of controllability, with low perceptions of control over threat linked to anxiety and depressive disorders (Cheng et al., 2013). How people tend to appraise controllability could impact whether they act to alter their emotion. Relatedly, people vary in the extent to which they tend to rely on model-based reasoning (Gillan et al., 2016; Patzelt et al., 2019), and trait affect-based impulsivity is associated with lower model-based reasoning – independent of a person’s current emotional state (Patzelt et al., 2019). This deficit in model-based reasoning could compound the effect of negative emotions on action selection, perhaps via the mechanisms we proposed above. Third, our account emphasizes that affect-related increases in GCs lead to heightened DA reactivity in the mesolimbic reward circuit (Figure 2). There are between-person differences in GR sensitivity to GCs, reflecting effects of genetics and chronic stress (Kosten et al., 2002; Ortiz et al., 1996; Rougé- Pont et al., 1993), and this sensitivity impacts the potency of GCs on DA reactivity (P. V. Piazza & Le Moal, 1996). A natural next step in developing our account would be to examine whether GR sensitivity is related to trait affect-based impulsivity.

We appreciate that we may not have made our focus on within-person effects sufficiently clear throughout the paper. We now clarify in the introduction that our focus is on how negative emotions shape computational processes, relative to that person’s baseline (pg. 5).

## Reviewer H

• On page 4 the modelling approach is described as ‘new’, but it draws heavily on existing accounts, many of which have been well rehearsed in the literature. The models in question are explanatorily powerful, meaning they can be applied widely as other authors have found.

## Author’s response

You are correct that decision neuroscience is by no means a ‘new’ venture, and you are right that our model draws heavily on existing work. We nonetheless believe our proposed model of affect-based impulsivity is novel, in that it synthesizes literature from disparate fields including preclinical models of addiction, ethological models of threat responses, affective neuroscience, and reinforcement learning. We have amended the text in the manuscript accordingly:

Here, we present a narrative review of affect-based impulsivity, propose an integrative decision neuroscience account of affect-based impulsivity, and outline a research agenda for testing central components of our account.

## Reviewer H

• On page 5, the authors ‘appreciate’ the value of cognitive neuroscience, and thankfully proceed to cite numerous such studies. Animal and computational models have great value in building accounts of impulsivity, but ideally we would want human cognitive neuroscience also to have a central role within a translational research program of impulsivity.

## Author’s response

We agree that human cognitive neuroscience has a central role in research on impulsivity, and we apologize that we seemed to have implied otherwise. As you point out, we cite human cognitive neuroscience studies throughout the manuscript. Moreover, the proposed model of affect-based impulsivity and corresponding hypotheses are designed to be tested using human neuroscience methods, as highlighted on pg. 19 and pg. 22-23 of the manuscript.

## Reviewer H

• It isn’t counter-intuitive that aversive states should promote pursuit of appetitive cues - mood repair is obvious and well established within the addiction literature (e.g. by Koob and others).

## Author’s response

We agree that it is well-established that negative emotions can lead to the pursuit of appetitive cues. Yet, from the perspective of action tendencies, the decision to pursue rewards when in duress is a bit surprising in that aversive states typically promote inhibition. We have thus revised this sentence to clarify the vantage point for our comment:

Given that negative emotions are often associated with inaction and withdrawal (e.g., de Berker et al., 2016), it is potentially counterintuitive that aversive internal states would promote pursuit of appetitive cues.

## Reviewer H

• Page 12 - serotonin doesn’t inhibit behavior, at least not universally. SSRIs increase escape behavior on a forced swim test. A citation is needed for DA increasing approach behavior when the outcome is perceived as controllable - it is true that DA does interact with contingency but approach behavior could simply mean approaching Pavlovian cues.

## Author’s response

We appreciate that our original wording oversimplified the complex interaction between serotonin and DA, as well as the role of each in behavior during aversive contexts. To provide additional nuance, we have added a footnote on pg. 13:

Historically, DA has been thought to guide approach behavior in appetitive contexts and serotonin to inhibit punished behaviors^3^ in aversive contexts (Boureau & Dayan, 2011).

^3^Though serotonin is most consistently implicated in behavioral inhibition, this effect is not universal. For example, SSRIs can increase escape behaviors during a forced swim test in rats (Detke et al., 1995).

We apologize for not including the citation for the role of DA in escape behaviors. We have amended that sentence accordingly (pg. 14):

Crucially, these DA-dependent behaviors only occur when the outcome is perceived as controllable (for review, see Boureau & Dayan, 2011).

## Reviewer H

• On page 19, the authors claim that their model does not address the role of habit, yet discuss cached values and a model-free system. It is possible to differentiate model free and habitual systems, but ultimately these terms arise from different paradigms but refer to something broadly similar. Overall, the authors should choose whether they are committing to a rather specific language in which e.g. model-free learning and habit can been distinguished, or a rather loose language in which ‘Pavlovian learning’ is unitary.

## Author’s response

You are correct that habit and model-free systems are tightly intertwined, with habitual action reflecting the use of cached values to make a current decision. Nonetheless, we are hesitant to claim that our model addresses the role of habit, particularly in light of the proposed neurobiological targets in our model (NAcc, mPFC). Given the anatomical dissociation in circuits that govern habitual versus goal-directed control, with the dorsolateral striatum supporting habit and dorsomedial striatum involved in goal-directed action, greater consideration of how these circuits interact with NAcc and mPFC is warranted. Moreover, the paradigms that are used to assess whether an action is under habitual or goal-directed control often differ, with devaluation procedures being central to studies of habit. As such, when considering concrete next steps for testing this account, there is a need to consider experimental methods that are wellsuited for disentangling habit and goal-directed action, as well as Pavlovian influences.

We agree that clarifying the role of habit is an important future direction, especially because of model-free learning processes may underlie habitual behavior. As such, we have amended the following sentences on pp. 21:

Third, negative affect may increase habitual behavior (Schwabe et al., 2010, 2012; Schwabe & Wolf, 2009, 2013), yet our model does not directly address the role of habit. Notably, habitual control relies on model-free learning systems (for review, see Dolan & Dayan, 2013). Thus, it may be fruitful to extend our account to consider more habitual impulsive behaviors. Such an extension would require further consideration of how repeated experiences consolidate model-free representations into inflexible stimulus-response policies, and whether this process is dependent on affect-based impulsivity.

## Reviewer H

• The review doesn’t appear to distinguish between impulsive action or choice, and within choice, risk taking and delay discounting. Again, this may be as a result of a desire for internal consistency with the cost of generalizability/scope.

## Author’s response

Thank you for pointing out this issue. You are correct that our manuscript, as originally written, did not delineate between different types of impulsivity, nor describe which are most relevant to our model. As such, we have added the following text to the introduction on pg. 5:

Second, impulsivity is not a unitary construct, referring to distinct behavioral tendencies that range from heightened delay discounting to inhibitory control deficits to impulsive decision-making style (Caswell et al., 2015; Sharma et al., 2014). In our model, we focus on impulsive behaviors that are short-sighted and rash, reflecting a preference for immediate rewards, consistent with original psychological theories on trait affect-based impulsivity (Cyders & Smith, 2008; Smith et al., 2007).

References:

Hall, N. T., Schreiber, A. M., Allen, T. A., & Hallquist, M. N. (2021). Disentangling cognitive processes in externalizing psychopathology using drift diffusion modeling: Antagonism, but not disinhibition, is associated with poor cognitive control. *Journal of Personality*, 89(5), 970-985.

Schreiber, A. M., Hall, N. T., Parr, D. F., & Hallquist, M. N. (2025). Impulsive adolescents exhibit inefficient processing and a low decision threshold when decoding facial expressions of emotions. *Psychological Medicine*, 55, e105.

Sripada, C., & Weigard, A. (2021). Impaired evidence accumulation as a transdiagnostic vulnerability factor in psychopathology. *Frontiers in psychiatry*, 12, 627179.

Weigard, A., Clark, D. A., & Sripada, C. (2021). Cognitive efficiency beats top-down control as a reliable individual difference dimension relevant to selfcontrol. *Cognition*, 215, 104818.